# The Influence of Smart Manufacturing towards Energy Conservation: A Review

**Shane Terry [1], Hao Lu [1], Ismail Fidan [2],\* 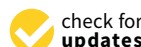, Yunbo Zhang [3], Khalid Tantawi [4] , Terry Guo [5] and Bahram Asiabanpour [6]**

[1]  Department of Mechanical Engineering, Tennessee Tech University, Cookeville, TN 38505, USA;
    SMTerry42@students.tntech.edu (S.T.); HLu42@students.tntech.edu (H.L.)

[2]  Department of Manufacturing and Engineering Technology, Tennessee Tech University,
    Cookeville, TN 38505, USA

[3]  Department of Industrial and Systems Engineering, Rochester Institute of Technology,
    Rochester, NY 14623, USA; ywzeie@rit.edu

[4]  Department of Engineering Technology and Management, University of Tennessee at Chattanooga,
    Chattanooga, TN 37403, USA; khalid-tantawi@utc.edu

[5]  Center for Manufacturing Research, Tennessee Tech University, Cookeville, TN 38505, USA;
    nguo@tntech.edu

[6]  Department of Manufacturing Engineering, Texas State University, San Marcos, TX 78666, USA;
    ba13@txstate.edu

\*  Correspondence: ifidan@tntech.edu; Tel.: +1-931-372-6298

**Abstract:** Today, the current trends of manufacturing are towards the adaptation and implementation of smart manufacturing, which is a new initiative to turn the traditional factories into profitable innovation facilities. However, the concept and technologies are still in a state of infancy, since many manufacturers around the world are not fully knowledgeable about the benefits of smart manufacturing compared to their current practices. This article reviews several aspects of smart manufacturing and introduces its advantages in terms of energy-saving and production efficiency. This article also points out that some areas need further research so that smart manufacturing can be shaped better.

**Keywords:** smart manufacturing; energy; cyber-physical system; data analytics; Industrial Internet-of-Things; artificial intelligence; additive manufacturing; robotics

## 1. Introduction

### 1.1. Introduction to Smart Manufacturing

Smart manufacturing (SM) is a production system integrated by multiple subsystems for data exchange, through an interconnected network. It allows production to change quickly based on a more complex array of factors. Through data analysis and decision-making, the process can be better tailored to meet the requirements of production, affording the user better control of the quality and superior optimization in the overall process.

In 2015, SM was first defined in the United States in Congressional Bill S.1054 as "a set of advanced sensing, instrumentation, monitoring, controls, and process optimization technologies and practices that merge information and communication technologies with the manufacturing environment for the real-time management of energy, productivity, and costs across factories and companies" [1]. The National Institute of Standards and Technology (NIST) defines SM as "fully-integrated, collaborative manufacturing systems that respond in real-time to meet changing demands and conditions in the

factory, in the supply network, and in customer needs" [2]. As represented in Figure 1, SM is the combination of advanced technologies in an interconnected manner to improve efficiency.

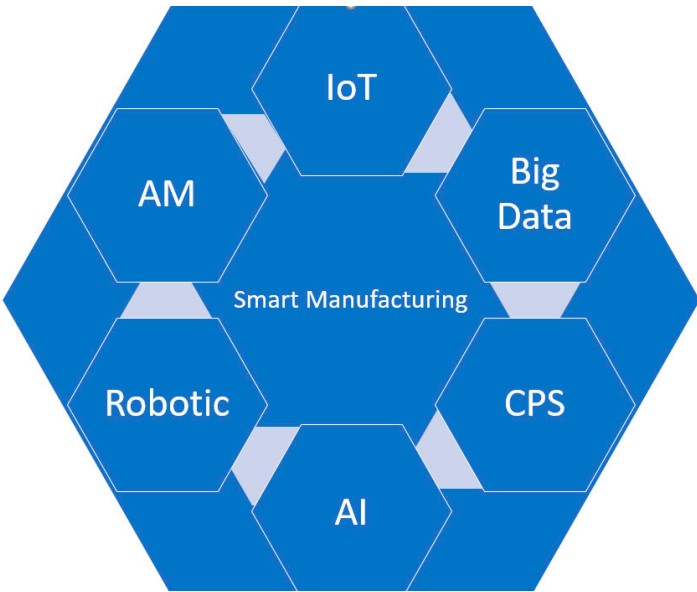

**Figure 1.** Representation of the combined technologies, such as Internet of Things (IoT), Big Data, Additive Manufacturing (AM), Artificial Intelligence (AI), Cyber Physical Systems (CPSs), and robotic technologies included in SM.

In 2010, China, for the first time, surpassed the United States as the world's largest manufacturer, and according to current data, continues to widen its lead [3]. Furthermore, China takes the global lead in its push towards industrial robotics built with artificial intelligence (AI) and places SM as a priority for its growth in China's industrial policy named "Made in China 2025", as stated by China's President Xi Jinping [4]. In the following years, the European Union and Germany adopted plans that largely rely on SM technologies to increase their national manufacturing production, and as a result, the new term "Industry 4.0" emerged to define the fourth generation of industry [5,6]. *The Wall Street Journal* referred to it as the "New Industrial Revolution" in 2013, and *The Huffington Post* called it a leaving "bullet train" that will "propel the manufacturers that climb on board" [5]. In Korea, the Manufacturing Industry Innovation 3.0 strategy was launched in 2014 to introduce SM.

The Smart Manufacturing Leadership Coalition (SMLC) states that "SM is the ability to solve existing and future problems via an open infrastructure that allows solutions to be implemented at the speed of business while creating value-added results" [6].

*1.2. Cyber-Physical System (CPS)*

1.2.1. Cyber-Physical Systems and Smart Manufacturing Trends in Advanced Manufacturing

Cyber-physical systems is the concept of traditional cybernetics via the Internet and the Internet of Things. It combines control, computing, and communication to make the physical world and the computer virtual world seamlessly integrated. The goal is to achieve network-based distributed real-time control. Cyber-physical manufacturing systems (CPMS) and SM applications in advanced manufacturing are growing trends. Cheng, J., et al. have investigated the role of 5G and Industrial IoT (IIoT) advancements in manufacturing [7]. The research group proposed the architecture and implementation methods of 5G-based IIoT and 5G applications, including enhancing mobile broadband (eMBB), massive machine type communication (mMTC), and ultra-reliable and low latency communication (URLLC) in different advanced manufacturing scenarios and technologies.

Monostori L. et al. outlined research directions for the implementation of Cyber-physical production systems (CPPS) [8]. The research group identified the three main components of CPPS as intelligence, connectedness, and responsiveness. In this research, the convergence of virtual and physical worlds and their elements were mapped. Furthermore, the changes that occurred due to the transformation from the automation hierarchy to a CPS-based automation were recorded.

Lee, J. et al., developed a CPS architecture for the Industry 4.0-based manufacturing system where their proposed 5C (connection, conversion, cyber, cognition, and configure) model provides a guideline for CPS implementation in manufacturing applications [9].

Kim and Park suggested a CPS-based manufacturing system optimization strategy [10]. Roger Burger identifies additive manufacturing (AM), sensors, autonomous vehicles, robots, advanced materials, and advanced manufacturing systems as the six main elements of a Factory 4.0 model [11].

Overall, several of these reported studies provided valuable information resources about the benefits of having CPPSs in their overall production yield and energy saving through production gains.

### 1.2.2. Energy and Cost Saving in Cyber-Physical Systems

All green technologies aim to conserve energy and natural resources, while maintaining or improving human living standards. Many efforts in achieving green information technology (IT) have been in the area of cyber-physical computing [12]. It is believed that new applications of computing go beyond the traditional applications of IT and now include smart software/hardware/network to monitor and control the systems in a human-free fashion [13]. Hahanov et al. listed "10 commitments of Sustainable Green Computing Development" and proposed a computing in sustainable development model, in which they grouped as reflecting/monitoring of the physical processes through singular (single computing), network (network computing) and global computing (global computing—Internet); controlling/managing of the physical processes through cloud computing, cyber-physical networks, and the Internet of Things; creating intelligent cyber-physical processes through brain computing and smart big data networks (Figure 2) [13].

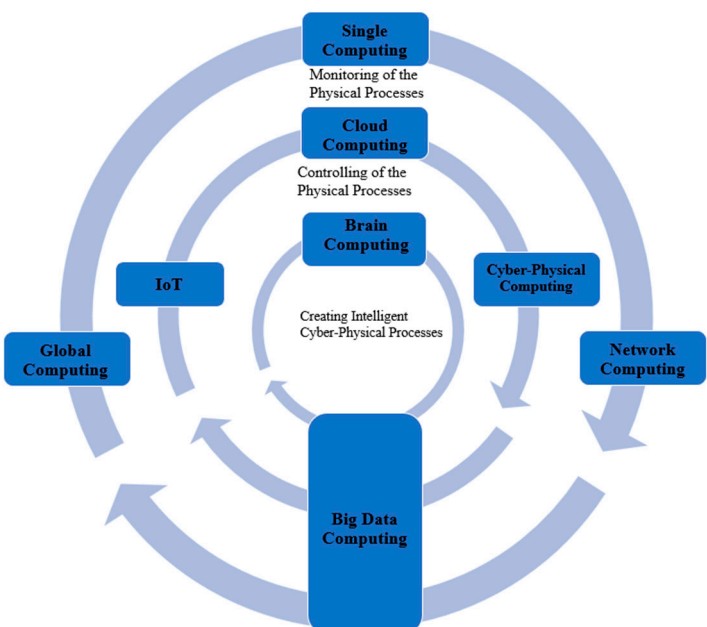

**Figure 2.** Hierarchy of the sustainable computing framework, and the lifecycle of the overall processes [13].

According to Nikolaos Doukas, the benefit of IoT is that large amounts of data (i.e., big data) are collected and processed. Unlike many other fields, energy efficient/green computing is not necessarily achieved through competing against system performance, as it can be achieved through

the development of efficient algorithms [14]. It is shown that dynamic power dissipation is the main source of power consumption in a digital system and system-level optimization has the most significant energy-saving effects [15].

Energy saving in CPS systems is not limited to software optimization. Siti et al. showed energy saving in the usage of 4th Gen versus 3rd Gen Intel® Core™ Processors. They found that the 4th Gen's Fully Integrated Voltage Regulator (FIVR) reduces energy consumption in the CPU and operates in lower temperatures [16].

*1.3. Traditional versus Smart Manufacturing*

SM is a production system integrated by multiple subsystems for data exchange through the Internet. It allows production to change quickly based on supply and demand. Through data analysis and decision-making, the company can better meet the current market demand, and the production can be better improved and tailored as necessary, lending to a more refined quality control system. The American SM system needs a unified standard to ensure the normal operation of each subsystem. These integrated standards span the three main manufacturing cycle phases: product, production system, and business. The NIST openly expressed the distress of manufacturers, indicating that the growing demand is experience in the following ways: more diverse customized services, smaller production batches, unpredictable supply chain changes, and disruptions. Successful manufacturers have to adapt to rapid changes while improving product quality, by optimizing energy and resource usage. Due to the increased implementation of smart devices in manufacturing, the amount of data feedback is also constantly expanding. With large amounts of data at the core of manufacturing, the question becomes "How to maximize the use of the data?" [17]. This core can achieve the maximum flow of enterprise data; more importantly, the data can be reused throughout the enterprise. However, communication among different heterogeneous systems can only rely on standards. The information standards given by NIST can satisfy the three manufacturing lifecycle dimensions: product, production system, and business [18]. In 2014, the President's Council of Advisors on Science and Technology (PCAST) issued a report in which it identified three priority manufacturing technology changes: Advanced Sensing (AS), Control & Platforms for Manufacturing (ASCPM), and Advanced Materials Manufacturing (AMM) [19]. AS and ASCPM increase the ability of manufacturers to respond efficiently and quickly based on the feedback information. Only standard systems can provide this reliance on effective information flow and fast system response capabilities. At the end of the report, the committee also pointed out that the standard systems can stimulate the adoption of new technologies, new products, and new manufacturing methods [19]. The core features of SM include the following: a comprehensive digital manufacturing enterprise with interoperability and enhanced productivity; real-time control and small-batch flexible production through device interconnection and distributed intelligence; coordinated supply chain management that responds quickly to market changes and supply chain imbalances; integrated and optimized decision support to improve energy and resource use efficiency; and the achievement of high-speed innovation cycles through advanced sensors and data analysis technologies throughout the product cycle [20,21].

SM is a production system that goes beyond the factory floor by implementing cyber-physical intelligent systems through a dynamic response time, that allows the system to better adapt the manufacturing process to specific product and energy needs. It involves automated control, integrated manufacturing, and networked companies improving productivity through information sharing and informed decision-making. SM provides the right information at the right time to the user in an understandable manner. There are the following levels to this integration of automation: manual, reactive, programmable, variable, and intelligent controls. At each concurrent level, there is more potential to save energy, and there may not be a direct reduction in energy cost. However, the greater potential stems from a greater ability to tune the system for higher efficiency. These levels for SM are represented stepwise in Figure 3.

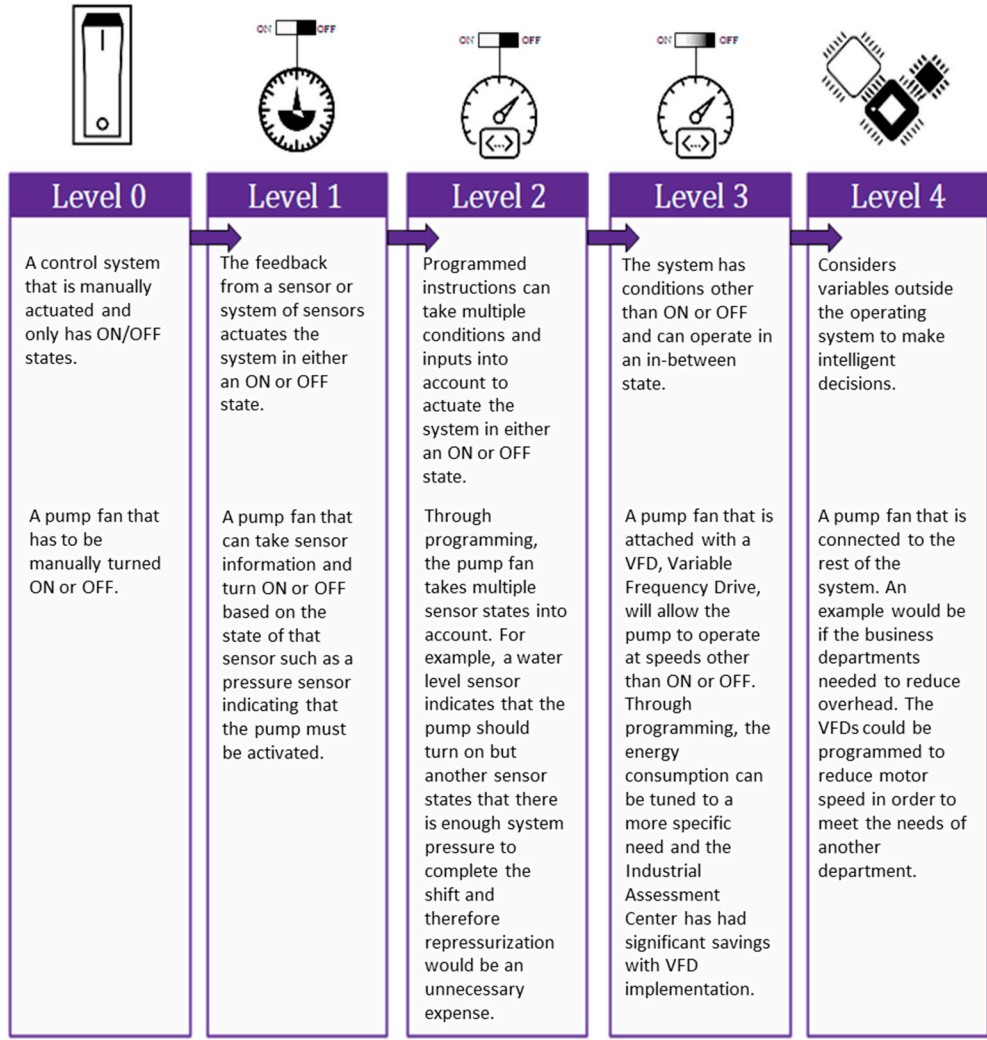

**Figure 3.** Demonstration of the levels for SM with a pump fan example of increased system complexity [22,23].

The difference between traditional manufacturing and this smarter implementation is an integration vertically through the production line and horizontally across departments and systems. A smart phone enables functionality beyond that of a traditional phone, by incorporating multiple functions in a single device. Similarly, SM enables greater customization and optimization, through the use of smarter processes to connect subsystems to a wider operational grid.

Through the implementation of SM, the ultimate goal is to handle information only once, enabling optimization of assets, synchronization of enterprise resources, supply-chain resources, and automation of business processing response to customer demands [24].

*1.4. NIST*

NIST is a US-based department that functions to adjust and publish the standards on various fields of technological research. NIST has broad research ranges, such as biotechnology, chemistry, semiconductor electronics, ceramics, physics, and optoelectronics. Countless products and services depend on the technologies, measurements, and standards provided by NIST. The main competency of NIST includes three aspects: measurement science, strict traceability, and the development and use of standards. NIST filed Current Standards Landscape for Smart Manufacturing Systems in 2016. In this article, the SM ecosystem is used as a basis to classify and evaluate existing standards and identify new standard active areas that can facilitate the implementation of SM systems. On the way to achieve

SM, the classical manufacturing system must be properly integrated. The evolution of using smart devices in multiple facets of the manufacturing process has allowed for the following innovations: embedded intelligence at all levels, predictive analytics, and cloud computing technologies [18,25]. All of these technologies will rely on a unified set of standards. For example, the standard for materials characterization methods, process metrology, sensing, control methods, algorithms, and information system frames can help users quickly choose materials and processes for AM and simplify design-to-product conversion [26]. Many organizations developing standards are international in their scope and functioning. As an example, International Organization for Standardization (ISO) also develops and supports a high number of standardization works on related concepts, from the energy efficiency of heat pumps to AM.

### 1.5. Smart Manufacturing Standardization Efforts

The international standard development organizations (SDOs) push for the advancement of the SM standards in a diverse array of fields on an international level. An example of some of the SDOs that have furthered the development of SM are the following: the aforementioned ISO, the International Society of Automation (ISA), Institute of Electrical and Electronics Engineers (IEEE), International Telecommunication Union Telecommunication Standardization Sector (ITU-T), and the International Electrotechnical Commission (IEC). Each of these SDOs and many more have contributed to the development of SM standardization. With the advancement of standardizing the process and procedures of SM, the technologies can more easily be invested in by the governing bodies leading to greater growth in the developing field.

The concept of SM has recently been widely invested in by the United States, Korean, and Japanese governments, by dedicating a significant amount of funding into the newly developing area of study. With the many benefits that come with SM implementation, numerous countries in Europe have seen significant savings, by adopting variations of Germany's "Industry 4.0" standard. Further adoption will revolutionize the current manufacturing field with changes in mass customization, waste reduction, and utility savings. The technologies that enable SM are made up of the following components: intelligent automation, IoT, AM, augmented reality, big data analytics, automated simulations, and cloud computing. The overarching concept of implementation is seamlessly combining these technologies and integrating them into a collaborative system. The goal of manufacturers is to reduce overhead costs without sacrificing product quality and production time. One of the most effective methods of overhead reduction is increasing system-wide energy efficiency. It is also worth noting that the energy savings potential that stems from inefficient industrial consumption makes up several billion dollars per year possible savings in the U.S. alone [27]. These statistics demonstrate the availability of savings afforded to the manufacturing sector if energy efficiency is improved.

## 2. Materials and Methods

### 2.1. Introduction to Data Analytics

The increasing complexity related to the optimization of the manufacturing process requires the increased use of smart devices on a series of connected subnetworks exchanging vast amounts of data. In this case, data analytics becomes very important. Data analysis is the use of statistical analysis methods to analyze the collected data, summarize and understand the data, and maximize the role of data. Data analysis was mathematically established in the early 20th century, but it is further developed and promoted with the advancement of computer technology. With the development of the Internet, IoT, big data, cloud computing, AI, and other new-generation information technology, it has brought valuable opportunities to many industries [28]. SM and data analytics have a potential for today's manufacturing industry, to solve energy efficiency concerns from the equipment level to the entire manufacturing plant. With the intelligent communication systems that they provide, it is proven that both can easily make manufacturing industries more productive, affordable and competitive.

The huge potential value of analyzing big data has attracted attention in many fields. SM is one of them [29]. The purpose of industrial big data is not only to pursue a large amount of data replacement through systematic data collection and analysis, but also to maximize the values. Big data is used to solve and avoid "invisible" problems in manufacturing systems and achieve a worry-free manufacturing environment. It can also be used to provide intelligent value-added services to the users of products [30]. Proper use of "Big Data" and the analytics to interpret the information provides supportive decision making and opportunities for manufacturers to stay competitive [31,32].

### 2.1.1. Digital Thread and Digital Twin

Digital twin is a digital replica of a physical product. Digital thread is an information bridge between digital twin and physical products. Digital twin is the embodiment of CPS. The concept of digital twin was first presented by Grivese in 2002 [33]; it was developed to aid the design of machinery [34]. Digital twin is a concept related to the previously mentioned CPS, but differs in some key aspects. Digital twin represents a physical product digitally, which can help simulate what should happen on the actual physical "twin." Digital twin is a prerequisite for CPS [35], and is directly represented by the technologies of augmented reality and virtual reality within SM.

According to digital thread, all data models can communicate in both directions. Thus, the status and parameters of real physical products will be sent back to the digital model through the CPS integrated with the smart production system. This will result in a consistent digital model for all links in the cycle and the ability to implement dynamic and real-time assessment of the current and future functions and performance of the system. During the operation of the equipment, the continuous and increasing data collected by the sensors and machines are used to interpret and utilize the data. These data points can integrate the requirements of later product manufacturing, operation, and maintenance into the early product design process and form design improvements, as Figure 4 shows. Digital twin is not just a simulation, but is also used for monitoring, control, diagnostics and prediction [36]. This is due to the bidirectional nature of the digital thread connection to the digital twin, where neither the physical nor the digital representation are technically the simulation of the other. Rather, the factors that influence the physical model influence the digital, and vice-versa.

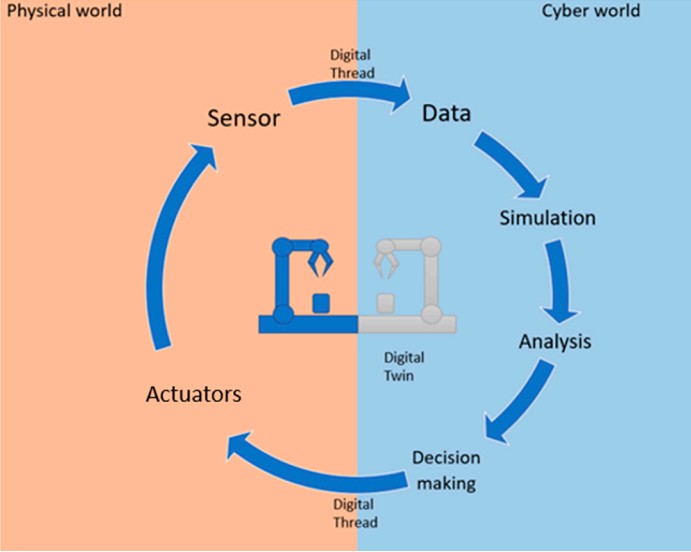

**Figure 4.** Digital Twin.

However, merely establishing a full-machine finite element model cannot be called digital twin. It requires that all real manufacturing dimensions can be sent as feedback to the model in production. Digital twin describes a model of each specific link connected through digital thread.

Digital thread is the result of integrating various links and then incorporating them into SM systems, digital measurement inspection systems, and CPS. Integrated models of the entire process through digital thread can further help the user to determine what will happen on the actual physical products eventually [37,38]. The concept of digital twin has attracted a lot of industrial interest in the past few years. GE's digital twin implementation saved $360 K by predicting a power outage in a gas plant [39]. Nanyang Technological University (NTU) in Singapore uses the Digital Twin technology and uncovered campus-wide energy savings of 31 percent [40]. Microsoft is also working on the Azure Digital Twins; it can model the relationships and interactions between people and devices [41].

Despite the significant investment of major tech companies, the concept and implementation of Digital twin is limited in the following ways:

- Digital twin's current focus is mostly on operation and maintenance.
- There is a lack of reference models.
- The research questions and challenges of digital twin are superficial [37].

With the advancements in IoT technology, the popularity of the digital twin is increasingly growing. Now, a sensor could easily be attached to a physical object and operational data could be remotely collected precisely and then, the object could be smoothly controlled from its digital twin. Creating a digital twin could be divided into several steps. Selecting a technology that helps your need in real time data flow from the IoT device is the first step. The next step is your decision on the operation of your digital twin. In monitoring your system, it is your main task to track the operational performance. Overall, your implementations could start small, but grow over time.

### 2.1.2. IIOT in Smart Manufacturing

The ongoing research into the optimization and minimization of energy usage has led to the development of many strategies to achieve efficient use of energy. As the room for improving energy efficiency becomes smaller and smaller, a systematic optimization mindset should be considered to increase the efficiency further, without relying on revolutionary inventions. It is now possible to make better use of massive data collected in real-time from a large number of sensors attached to CPSs, thanks to the advancement of technologies. These emerging technologies, including IoT or IIoT and big data analytics [42–49], make the overall system processes more available for optimization, leading to less energy waste. From a system perspective, when all individual units run harmoniously by following some optimized rules, energy waste is minimized. This can be thought of as the "smartness", i.e., given a condition, an actuator acts by the following instruction. IoT can provide the required conditions, while big data analytics can offer the instructions. At a higher viewpoint, the objective of industrial energy efficiency is aligned with Industry 4.0, or a SM framework. Figure 5 shows a conceptual diagram of such an "optimal" system, with feedback control aided by IIoT and big data analytics.

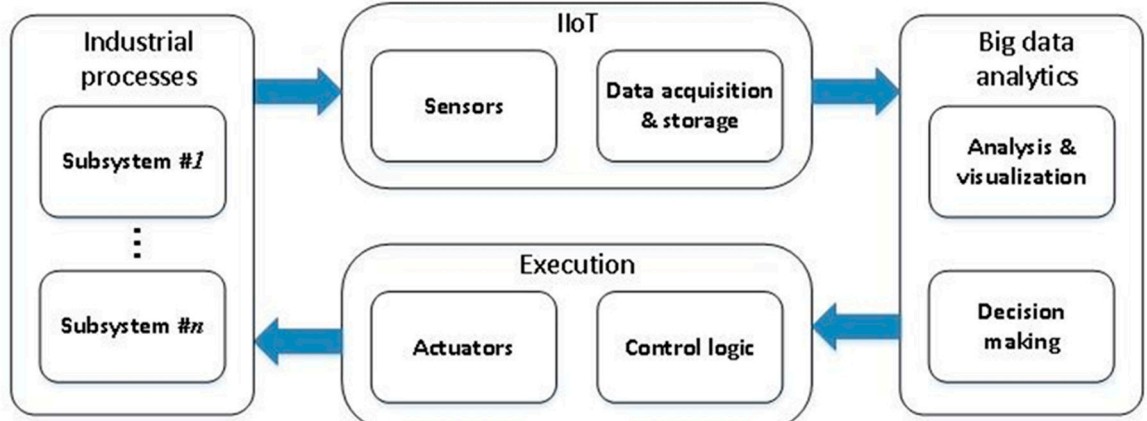

**Figure 5.** System with feedback control aided by IIoT and big data analytics for energy conservation.

Tan et al. [50] proposed an IoT enabled software application for the real-time monitoring of energy efficiency in manufacturing shop floors. In particular, the authors introduce an approach that uses both the energy and production data for energy efficiency assessment. The research applies the data envelopment analysis (DEA) technique, to identify abnormal energy consumption patterns and quantify energy efficiency gaps. Finally, the author provides quantitative evaluation results via a case study [51].

Yan Li et al. [52] introduced a functional framework of an IoT-based energy management system (EMS) and proposed a set of evaluation indices for industrial energy-intensive equipment. The authors argue that the existing evaluation methods cannot comprehensively reflect the energy-consumption level of equipment, and it is demanding to develop evaluation index systems for industrial energy conservation and emissions reduction. Based on the IoT-based EMS and their proposed index system, the authors analyze an integrated method for a comprehensive evaluation of the energy consumption, where the method combines the analytic hierarchy process (AHP) [53] and a fuzzy evaluation technique.

Sachin Nimbalkar et al. examined SM technologies and data analytics approaches for improving energy efficiency and reducing energy costs in process-supporting energy systems; specifically, those with motors and drives, fans, pumps, air compressors, steam, and process heating. The authors explain smart technologies from levels 0 to 3 that can be applied to increase the process efficiency and introduced several technology vendors offering SM products, such as industrial automation devices, internet-connected sensors, comprehensive digitalization, monitoring and control platforms, etc. By using three real-world examples of industrial applications of smart technologies and data analytics, they also explain how SM and IIoT are already affecting different business sectors. According to the authors, the most energy-efficient technologies implemented from the DOE IAC1 and ESA2 assessments are level 0 or 1 (simple controls or control devices with sensors), and implementing higher-levels of smart technologies (level 2 or 3) is likely to increase energy savings and could make energy savings sustainable [54].

Yingfeng Zhang et al. proposed a big data-driven analytical framework (BDDAF) for reducing the energy consumption and emission for energy-intensive manufacturing industries (EIMIs). The research is interdisciplinary, including manufacturing, energy, and big data. In the proposed framework, energy big data analytics is implemented based on two key components: energy big data acquisition and energy big data mining. The author implies that traditional EIMIs are in harsh production environments, and the IoT technology and soft sensor approaches can help collect the multi-source and heterogeneous energy big data. In the analytics aspect, it is proposed that a closed-loop structure of energy big data mining could mine valuable knowledge and patterns from the big data in the EIMIs. In the paper, an application scenario is presented to show the effectiveness of the proposed framework. It claims that the energy consumption and energy costs are reduced by 3% and 4% respectively [55].

Recent research reported by Nader Mohamed et al. [56] is about how smart techniques can enable opportunities for energy efficiency improvement in smart factories by using a three-layer enabling architecture in Industry 4.0 framework [53]. The architecture follows the six design principles of Industry 4.0 [57] and consists of a CPS manufacturing services layer, a fog manufacturing services layer, and a cloud manufacturing services layer [54]. This architecture is based on several services available on various technologies, such as IIoT and/or Internet of Services (IoS), manufacturing CPS, fog computing, and cloud computing. The authors suggested that these services should be integrated using a service-oriented middleware platform specifically designed for Industry 4.0 applications, and believed Man4Ware [58], one of such platforms, is a suitable option [55]. In particular, a blockchain-based service-oriented middleware was considered to support trusted information exchanges, automated and efficient negotiation processes, and efficient smart agreements among enterprises. Finally, a quantitative benefit analysis and comparison were provided to illustrate how much energy savings are achieved by using the Industry 4.0 solution.

As it could easily be seen from these reported studies, adding the IIoTs can enable direct energy savings for the smart factory of today.

### 2.1.3. Digital Thread/Twin in Smart Manufacturing

Gökan points out that creating a finer granularity on the machine level is necessary for achieving energy-saving targets [59]. Digital twin is a digital shadow of a physical product. Through integration with external sensors, it reflects all the characteristics of the object from the micro to the macro and displays the evolution process of the lifecycle of a product. Not only the product, but also the production system and machine maintenance, need digital twin [4]. The realizations of the feedback from the real physical system to the digital model are the most important inspiration of digital twin. In the industrial field, instead of importing the data to the real physical world, now people try to fit everything that happened in the physical world back into the digital world. In this way, the coordination of the cyber and physical world can be ensured throughout the entire lifecycle. Various types of simulation, analysis, and even the application of AI-based digital models, can ensure its applicability to the real physical world [25,34,37]. Without digital twin and digital thread, the SM system cannot be implemented [5]. Manufacturers can get a clear picture of actual performance through the digital twin and improve their situational awareness and operational flexibility [37]. The equipment energy consumption is a large part of the total manufacturing energy consumption and can benefit greatly from the aforementioned smarter technologies [60].

Overall, the digital thread/twin technology helps the manufacturers with the information needed to form intelligent solutions for the reduced use of energy. Several studies reported before provide best practices and case studies, which could be adapted and implemented.

### 2.2. Data Analytics

### 2.2.1. Multi-Criteria Decision Making (MCDM)

Decision-makers often face several conflicting choices. To help people make the best decisions, scholars develop MCDM to construct preferences and determine the correct relative weights of criteria [61]. In the last two decades, the field of MCDM has been growing rapidly in response to every evolving technological field. MCDM refers to the decision-making of a set of limited or infinite solutions that conflict and are non-commensurable. It is one of the important contents of analytical decision theory. There are two main theoretical streams: multi-attribute decision-making (MADM) and multi-objective decision-making (MODM) [62]. The theories and methods of MCDM are widely used in many fields, such as engineering, business, and the military [63–65].

Compared with traditional evaluation methods, MCDM can evaluate, queue, and select multiple items. When dealing with a problem, each influencing factor will be processed based on the criteria of the project. The weighted value of each factor is calculated by the extracted and processed data. By using multiple decision methods with the power of modern computers, a dynamic analysis system with a powerful analysis machine is established. A multiple criteria decision is a decision that needs to consider two or more criteria simultaneously. If the target of an enterprise is to choose one of several products for production, the size of the profit, the availability of existing equipment, the adequacy of raw material supply, and other factors must be considered. MCDM has been used in many fields, and one such example is the utilization of green building materials (GBM). Building materials should cover all three pillars of sustainability (3P). The capability factor in all GBM choices is a multi-criteria decision problem, which is very suitable for MCDM [64].

### 2.2.2. Energy Savings

The contradiction between industrial development, energy saving, and emission reduction remains prominent. Energy usage will damage the environment to varying degrees in all stages of production, transportation, and consumption. Reducing production costs is a challenge that several companies have been working hard to solve. Production capacity lags and may cause the company to be eliminated. By using automation technologies, operations can increase productivity and reduce manpower [66]. Factory automation can significantly contribute to energy efficiency

in manufacturing [67]. Energy-saving and emission reduction technologies have made important contributions to sustainable development. Facing the increasingly constrained industrial resources and environment, focusing on energy conservation and emission reduction, and taking a green and low-carbon development path are important guarantees for sustainable development. Energy security guarantee capabilities and low-carbon energy sources such as nuclear energy [68], wind energy, and solar energy should be developed vigorously [69]. In the current era of sustainable development, for supporting the current expanding economic energy demand, the energy need increases continuously. To reduce the use of traditional non-renewable energy and release environmental damage, energy planning is required. The MCDM method has been successfully used in energy planning processes and is considered to be the most appropriate method for solving energy-related problems [63]. Current energy planning has become complex, with multiple benchmarks, including technical, social, economic, and environmental. The topographical constraints about naturally distributed renewable energy systems have become more complex. In this case, all possible conditions of MCDM (including technology, institution, country, standard, social, economic, and stakeholder) are sorted and weighted. Then, MCDM gives a ranked decision for the user/algorithm to pick [18,70]. As robotic technology improves, industrial robots' energy efficiency has been improved. Riazi et al.'s algorithm [71] has reduced up to 45% energy without changing the original path in a task. A group of researchers uses cloud technology to minimize industrial robot energy consumption, leading towards sustainable manufacturing [72]. Yin et al. propose a machine learning based trajectory planning method. The simulation demonstrated the feasibility of the method and the optimization of energy use [73].

### 2.2.3. Artificial Intelligence (AI): High Fidelity vs. Low Fidelity

Today's AI has gradually entered all aspects of human production and life. It is one of the hot research topics today. The government also increased investment in AI in the 2021 budget proposal [74]. AI is developed when a machine obtains specific knowledge by analyzing and arranging a large amount of data. AI uses the obtained knowledge to solve related problems [75]. AI has a wide range of applications in manufacturing [76]. AI is an important part of SM to make a precise and reliable decision. With the modern innovations that IA has offered in the fields of AM, robotics, and digital technologies, energy companies are now exploring the possibilities of incorporating IA to increase the more efficient utilization of energy. Machine learning (ML) has been applied all over the manufacturing process from progress and operation to fault detection and quality improvement. The data models are important for ML. The accuracy of the AI prediction depends on the fidelity of the data used in the model. There are low-fidelity data and high-fidelity data.

The low-fidelity data model uses approximations to simulate the system. The high-fidelity data model uses the data which is a close match to the real environment. Therefore, the result of the high-fidelity model will be much closer to the real-world response than the low-fidelity model. The low-fidelity data is much cheaper in terms of computation cost to obtain results when compared to high-fidelity data. It is possible to obtain a large amount of data to train the algorithm. The high-density data is mainly used to build electronic models for a thorough simulation analysis of production, to optimize production [75,77,78].

### *2.3. Additive Manufacturing (AM)*

### 2.3.1. Introduction of the Technology

AM also known as three-dimensional (3D) printing, is becoming a disruptive technology that has begun to reshape the field of design and manufacturing. AM has significantly changed the world by providing more flexibility to product design, less cost in producing complex shapes, less need for assembly, and less time and materials cost in production run [79]. With these new affordances, people have applied AM in different sectors, including medicine and health care, architecture, transportation, aerospace, education, art, and aesthetic design. After experiencing double-digits

for 18 of the past 27 years, the AM market is expected to grow to $21 billion by the year 2020 [80]. Compared to conventional manufacturing methods, AM shortens product development cycles, reduces manufacturing cost and lead time, increases the sustainability of manufacturing, enables the production of customized and personalized products, breaks the conventional supply chains, and introduces the prospect of new business models. AM has also inspired the development of the maker movement, by democratizing design and manufacturing [79].

The following table (Table 1) illustrates the various AM processes.

**Table 1.** Classification of AM processes by ASTM International [81].

| | Material Extrusion | Powder Bed Fusion | Vat Photopolymerization | Material Jetting | Binder Jetting | Sheet Lamination |
|---|---|---|---|---|---|---|
| **Technologies** | Fused Deposition Modeling, Contour Crafting | Select Laser Sintering, Direct Metal Laser Sintering, Selective Laser Melting, Electron Beam Melting | Stereolithography | Polyjet/ Inkjet Printing | Indirect Inkjet Printing | Laminated Object Manufacturing |
| **Materials** | Thermoplastic, Ceramic/Metal Pastes | Polymer/Metal/ Ceramic Powder | Photopolymer, Ceramic | Photopolymer, wax | Polymer/ Ceramic/ Metal Powder | Polymer/ Ceramic/Metal Film |
| **Energy** | Thermal Energy | Laser Beam, Electron Beam | Ultraviolet Laser | Thermal Energy, Photocuring | Thermal Energy | Laser Beam, Ultrasonic Vibration |

Based on different working principles and printable materials, the recent ASTM standard has classified the major AM systems into seven categories: (1) material extrusion, (2) powder bed fusion, (3) vat photopolymerization, (4) material jetting, (5) binder jetting, (6) sheet lamination, and (7) directed energy deposition.

Material extrusion: fused deposition modeling (FDM) created layers by mechanically extruding molten thermoplastic material (e.g., ABS or PLA) onto a substrate. While most of the extrusion systems process thermoplastic materials, there are new efforts that have been made in processing ceramic and metal pastes. The FDM process has gained popularity among DIY crowds, due to its inexpensiveness and flexibility.

Powder bed fusion: The powder bed fusion techniques adopt an energy beam (e.g., laser or electron beam) to selectively melt a power bed. As long as a layer is scanned, the next layer of powder is spread via a rolling mechanism. Afterward, the next layer of powder is fused to the previous layer. There are powder bed fusion systems taking polymer and metal powder as materials. The selective laser sintering (SLS) process is typically adopted to process polyamides and polymer composites. This process can also be adapted to create ceramic and metal melting blends. The resulting "green parts" require high-temperature sintering as the post-processing. Direct metal laser sintering (DMLS), selective laser melting (SLM) and electron beam melting (EBM) are the most popular metal powder bed fusion techniques.

Vat photopolymerization: the stereolithography method (SLA) utilizes an ultraviolet laser to selectively polymerize the UV curable resins to create a layer of solidified material. More layers are subsequently cured until the part is complete. The process is limited to photopolymers, because it relies on photopolymerization. Some newly developed processes can process ceramic components (e.g., alumina, zirconia, PZT) by suspending nanoparticles in the resin.

Material jetting: Inspired by the 2D ink-jet printing technology, material jetting processes deposit wax and/or a photopolymer droplet onto a substrate via drop-on-demand inkjetting in a drop-by-drop or continuous manner. The jetted droplets undergo a phase change via heating or photocuring. Recently, new attempts in direct inkjetting of nanoink suspensions of ceramics, metals, and semiconductors are made to create parts with additional functionalities.

Binder jetting: the binder jetting process selectively deposits a liquid polymer onto a powder bed. The droplet infiltrates the surface of the powder bed and forms a printed powder agglomerate primitive. Then, the powder recoating is done via powder spreading. After completing the whole part, post-processing is required to solidify the part. This process can process any powdered materials that

can be successfully spread and wet by the jetted binder. Different materials including metal, ceramic, foundry sand, and polymer, are processed using binder jetting.

Sheet lamination: laminated object manufacturing (LOM) is adopted in an AM system, which brings the benefits of low internal tension and fragility of the parts, high surface finish details, and lower material, machine, and process costs. The metal parts can be created by the sheet lamination AM process, by cutting, stacking, and gluing profiled metallic laminates. Ultrasonic welding has also been introduced to produce functional gradient metallic structures.

Directed energy deposition: direct energy deposition (DED) AM processes directly feed the metal powder or wire into the focal point of an energy beam, to generate a molten pool. DED has a multi-axis motion platform similar to the three-dimensional welding machines. The most commonly used energy sources include lasers and electron beams. The DED process can also add a coating or clad to the existing surfaces.

In all these technologies, it was proven that AM can be an excellent solution for the production needs of any business with minimal energy consumption, and selecting the appropriate AM technology for the customers' requirements on the basis of complexity, material, and batch size is the key for the successful outcomes.

### 2.3.2. Green Technology

As pointed out by a recent paper [82], AM tends to be overlooked in energy scenarios. According to the research, it is expected that a 5% to 27% reduction of the global energy demand is achievable with the adoption of AM in 2050 [82]. The largest effect of AM on energy efficiency comes from the high material efficiency of AM compared to the conventional manufacturing processes [10,68–74,82–88]. AM's nature of "adding" material instead of "subtracting" material achieves high material efficiency in applications such as fuel mixing heads and diffusion burners for automotive or aerospace parts, and injection molding dies [85]. These parts have a high "buy to fly" ratio, and therefore result in high material waste, complex operations, and high maintenance cost using conventional manufacturing processes. Whereas there are only minor differences in the energy consumption for the AM of different materials, the differences in the subtractive production are substantial [84]. According to Hettesheimer et al.'s research [84], the projection for automotive and aerospace industries indicated that the energy-saving potential from AM may equal roughly 0.4% of the final energy consumption in the German transport sector.

In the use phase, AM also plays an essential role in energy efficiency. It can facilitate the customized production of strong light-weight products, and it allows designs that were not possible with previous manufacturing techniques [87]. The adoption of lightweight parts on aircraft or cars can substantially reduce fuel consumption and greenhouse gas emissions [83]. In Huang et al.'s paper [83], it is expected that the estimated fleet-wide life-cycle primary energy savings will at most reach 70–173 million GJ/year in 2050, with cumulative savings of 1.2–2.8 billion GJ, and the associated cumulative GHG emission reductions were estimated at 92.1–215.0 million metric tons. Furthermore, it was shown through evaluations that the energy efficiency with several light-weight parts, used to their full potential, reduced airplane fuel consumption to 6.4%, with a material saving of 4050 tons/year of aluminum, 7600 tons/year of titanium and 8110 tons/year of nickel. Baumers et al.'s study [88] indicates that the light-weight parts made of Ti-6Al-4V achieve a 39% weight reduction compared to the original part, and 2.6 × reduction in greenhouse gas emission. If these parts are assembled for a JetA1 aircraft, through its lifetime, the fuel-saving is $22,000, while the fuel-saving becomes $880,000 for a Boeing 747 aircraft [88]. Verhoef et al. expect that a 5% to 25% energy saving can be made [79] in the aerospace sector globally, due to the adoption of light-weight parts made by AM. Additive manufacturing also enables the production of parts with better performance for energy saving, through the process of part consolidation, minimizing the reduction in strength due to the fabrication process. The use of additively manufactured SiC high-temperature components saves energy and emissions in the industrial heating

processes [10]. Additive-manufactured metallic-3D Ox-Ox CMC achieved cooling-flow savings in all hot turbine stages [12].

In addition, AM has also significantly changed other aspects of the lifecycle of the production, including supply chains, handling, process time, power consumption, emission, labor and maintenance [10,82,83,87,88]. Verhoef et al. found that a 4% to 21% energy saving in the construction sector is achievable due to the effects of AM on feedstock and transportation together with the adoption of parts made by AM [82]. Thomas et al. pointed out the increased automation in AM could lead to a more cost-effective production [87]. AM has a competitive advantage in defense and aviation. AM reduces multiple individual parts into one part. AM has the potential to reduce aircraft weight by 55% and increase the sustainability of the aerospace industry. This technology affords manufacturing the ability to minimize subtractive tooling, reduce part count through part consolidation, and minimize assembly time on low part count productions [89].

Several of the research and case studies reported above proved that additively manufactured products required 50% to 75% less energy than large-scale manufactured goods.

### 2.3.3. Sustainability

Sustainability is a balance of resources and benefits, without further harming the environment. This includes improving energy efficiency and material utilization to maximize the carrying capacity of resources and the environment to achieve long-term benefits. The utilization of raw materials and energy conservation is of great significance for sustainable development. Compared with traditional manufacturing technology, powder metallurgy can save 44% in energy. The material utilization rate of powder metallurgy is 95%, much higher than the traditional metal process. Emissions of toxic and hazardous substances have also been reduced during the manufacturing process [90]. The AM fabrication method, often called "rapid prototyping," is the depositing of feedstock layer-by-layer to create a 3D part. Notably, 3D printers can quickly produce complex physical models within a relatively short time, and the systems themselves are simpler and safer to operate than most traditional fabrication equipment.

- Rapid Prototyping—Rather than creating a faulty part out of expensive material, manufacturing a simpler and quicker prototype in which to test can greatly minimize waste.
- Cost Savings—The aforementioned waste can be costly in not only material expenses but for some applications, it can save energy expenses. For high-energy applications such as forging or casting, printing a test model before the finished part provides many benefits.
- Customization—The benefit of additive instead of subtractive manufacturing is being able to create previously unachievable shapes [91].

AM's main benefits are customization and waste reduction. However, it is worth noting that the energy consumption on traditional fused deposition modeling 3D printers is greater than that of traditional methods. For prototyping, this energy difference can be easily compensated through waste reduction. This does mean that the functional non-prototype parts are only valid, from an energy standpoint, in lightweight low part count applications, such as aerospace [92].

### 2.4. Robotics

#### 2.4.1. Industrial Automation and Robotics

Industrial robots have been around for many years. The application of industrial robots has reduced the demand for personnel in the industry. In recent years, the development of information technology has promoted the intelligent development of robots. The ability to collect and share vast quantities of data among machines is a new advancement that has great potential to increase efficiency and robotic capabilities. The use of IoT-aided robots in industrial plants and smart areas grants manufacturing the following benefits: access control in restricted sites, assist during panic and

danger episodes, predict and avoid dangerous situations, and manage equipment and instruments autonomously [93]. The IoT can also be applied for more efficient data collection from machines, allowing for it to be more useful and to aid people or organizations more effectively in streamlining bulk data into usable data [94]. The interconnectivity of industrial automation presents an unparalleled increase in productivity; however, the advancements require the following challenges to be addressed: the need for a secure and high bandwidth network to communicate with robots; the ability to analyze, communicate, and interact with other robots; and the ability to coordinate robots' activities to optimize their capabilities within the network [93].

The ability to automate tasks in manufacturing will lead to further changes in the role of people in the industry. As factories become more automated, humans will move into specialized roles, where their input is irreplaceable. Many would think this means the obsolescence of mid-tier assembly workers and their removal from the process. Some, however, would argue that this idea of assembly line work being routine is a misconception that undervalues experiential knowledge and the adaptability of humans [95]. This common line of thinking also encourages the idea that people can be easily replaced by robotic automation in any field perceived as having simple steps when complicated processes can abound. Sabine Pfeiffer refutes the commonly thought of a situation where robotic labor eliminates mid-tier workers, creating a wide skill gap at facilities, with only highly trained employees, like engineers, and minimally trained staff, like janitorial staff [95]. Instead, he offers a new perspective where mid-grade workers are still present, heading off problems before they can affect the product and making the production line run smoothly, arguing that, as automated systems become more complex, a new area of labor will appear to reduce the vulnerability of the structure and maintain it. This side grade of the labor force will prompt the creation of new interfaces that allow for more efficient interactions between workers and machines. The use of augmented reality with wearable smart glasses and touch interaction through mobile devices has been explored in the case of maintenance and other basic tasks on the production floor. The results of the tests highlight several benefits and shortcomings of both augmented reality and touch interactions [96]. The ability of workers to communicate with the automated environment around them is of utmost importance to maximize their ability to maintain the technologically advanced systems.

The cloud is still a relatively new invention that allows for unique applications of older and newer technologies. Benefits of this include access to large libraries of data, off-site computing power, shared processes for machine learning, and the use of collective human skills. In the future, the use of cloud infrastructure could lighten the load of emerging users by removing the need for highly specific hard or soft -wares, through several business models. Unique features implicit in the cloud offer further the potential benefits, such as reduced costs of maintenance [97]. An example of its applications with robots is the reduction of on-site equipment. Complex tasks, such as grasping current technology, require lots of complex equipment and take up space within the production facilities, whereas in cloud computing, the computing equipment is stored elsewhere and instructions for tasks can simply be downloaded, saving space and energy [98]. Cloud technology is not bereft of its complications, though. Current issues include the allocation of computing resources, where data storage will be held, the scheduling of real-time demands and processing performance, security for the cloud-based computing servers and users, and the assurance of quality to maintain efficiency [99,100].

### 2.4.2. Energy Efficiency of Robots

As indicated before SM and its impact on energy efficiency is one of the hot topics in today's advanced manufacturing [101]. Several industries try to find solutions to cut their energy consumption by improving their current automated systems with robotic cells and accessories. This solution is an easy step since the robotics and automation concepts are not new for several practitioners. The current trends indicate a big demand for industrial robots.

In 2016, for the first time, the electronics industry exceeded the automotive industry in the demand for industrial robotics in the Asian markets of China, Japan, and Korea. Worldwide, the electronics

sector's share of the robotics market rose steadily to 32% in 2017, almost equal to the automotive sector (33%) [102,103]. This ratio is quickly changing, and the electronics and metal industries are quickly becoming prominent robotic markets. This change indicates that sectors that have not been historical markets for industrial robotics are now adapting to this robotics revolution.

The International Federation of Robotics (IFR) executive summary for 2019 projects that China alone will account for about 48% of the top 10 markets, with a size almost double that in 2016. Furthermore, the four East Asian countries of China, Japan, Korea, and Taiwan will constitute approximately 75% of the market [103].

Historically, improvements in energy consumption in robotized industrial plants have been achieved through improvements in hardware, software, or a combination of both [104]. Improving energy efficiency through hardware improvement is achieved by proper selection of the robotic system [105], replacement of hardware components with more efficient components [106], or the addition of components for energy storage and recovery, such as flywheels [104,107]. Energy efficiency improvement through software has been achieved by optimizing the trajectory of the robotic arm or through improving operation schedules [104,108]. However, the main disadvantage in industrial robotic systems is that industrial robots have to be isolated, and only operate in highly controlled and deterministic environments for safety [109,110].

In the next generation of intelligent industrial robotics, the approach of improving energy consumption is radically changed, due to the unique features over current-generation robotics in the advanced sensory and perception systems, control algorithms, as well as the enhanced data processing capability [111].

Compared to current-generation industrial robots, the next-generation intelligent industrial robots are more dynamic and compact and equipped with an advanced computing capability, coupled with an advanced sensing and perception system to sense a human presence around them. Robotic systems work together to achieve energy efficiency improvement, by allowing the unit to work collaboratively side-by-side with human operators. This is a significant improvement in the operations of industrial robots. This collaboration can simplify troubleshooting processes and reduce costs significantly for manufacturers. Ground studies conducted at Toyota, Ford, and Mercedes Benz indicated that deploying co-bots resulted in increased production [112]. Furthermore, the National Institute of Standards predicts that intelligent robotics can save manufacturers at least $40.4 billion annually [109]. As a result, the IFR forecasts that collaborative robots will take the lead in the robotics industry in coming years [113].

2.4.3. Collaborative Robots

A requirement for SM autonomous production is that robotic operators must complete tasks without the need for isolation. SM encompasses a wide range of topics, including the emergence of new robotic technologies that are perfect for the new industrial revolution.

Robots are the physical representation of the CPSs, as mentioned previously. The introduction of robots into the manufacturing setting has propelled production far beyond what was previously achievable. To compare the 1990s to 2014 markets, the three largest companies had a market share of $36 billion to $1.09 trillion, with 1.06 million fewer employees. These were the three largest companies in Detroit (the 1990s), and the three largest companies in Silicon Valley (2014). The widespread introduction of robotics resulted in a necessary workforce that is 90% smaller. By connecting the robots system-wide, the results in Germany suggest another financial and productivity boost for the US manufacturing industry, by following similar methods. Today, several economists believe that robotics and automation represent a wave of technological change that could lead to a structural shift in the manufacturing industry [114].

This intelligent control scheme for operating robots affords the companies and operators greater control over the efficiency of automated manufacturing. A robotic technology that directly benefits is the principle of "Collaborative Robots." Commonly referred to as "Co-Bots," collaborative robots

are intelligently controlled systems that have feedback loops in place that safely allow a robot and operator or robot and robot to work in tandem.

The operation of co-bots varies from manufacturer to manufacturer; a common method of collaborative sensing is through the use of a sensory field, where, for a given space, the controller is aware of all objects, personnel, and features that could influence the operation. This field is generally done with cameras, laser detection, and/or thermal imaging. Another method of collaborative sensing is through the use of on-board sensors. Robots that are intended to work in tandem are equipped with safety sensors that slow or stop motion when touched by an object that introduces an increase of torque. With these sensors equipped, the controller will hard-stop when contact with an operator is made. For most robot manufacturers, this tolerance can be set so that injury can be minimized/avoided in the event of an accident.

In relevance to SM, the efficient use of co-bots requires a reliable, safe data stream. The prerequisites to maximizing the effectiveness of the robots require a majority of the SM technologies: big data analytics, IIoT, and digital twin and thread. It is the processing of all concurrently that allows for the most energy-saving potential.

## 3. Discussion

Many factories in manufacturing today are semi- or fully automated. Only a few employees are needed to operate the machines. However, this cannot be called SM. Smart manufacturing requires the entire industry chain to form an interconnected network comprised of subnetworks linked together. IoTs collect data from all links from supplier to production to business and import the data into a virtual data model for analysis, to further optimize and adjust each link to minimize or even avoid problems, energy, and material use. Large amounts of data need to be transferred and analyzed quickly. The fidelity of the data used to train the model also requires trade-offs to enable high-speed, accurate simulation and prediction. A better digital model is needed for dynamic real-time monitoring and the optimization of product quality, energy consumption, mechanical wear, etc. This poses some challenges for sensors, storage, data complexity, quality, and communication. Various devices are used in this process, and unified standards are required to achieve fast and effective data communication between the devices.

According to the definition of SM, data is centralized, linking the CPS to all subsystems to capture the data input, and the data must be processed and sent as an output non-locally to coordinate activities. The increase in interconnectivity requires increased cybersecurity and network speed to ensure that the processes that are connected are also secure. Sustainable development and energy conservation are the way forward. By analyzing data and optimizing the entire industry chain, most functions can be improved. However, data analysis also brings demand for processors with high energy consumption and high computing power.

There are two options for working with big data. One is to develop new hardware to increase computing power to achieve high speed. However, high computing power is directly proportional to energy consumption. The second is the development of new high-speed algorithms to reduce the need for hardware systems. The energy consumption of robots has declined in recent years. A lot of researchers show that robots can be further optimized for energy saving. The rapid development of technology has also made unified frameworks and standards necessary. In this way, new technologies can be used quickly.

It is very clear that manufacturing has a new face in the 21st century. The traditional method of manufacturing practices does not provide as much benefit when compared to the smarter technologies available to companies. Today, customer requirements mostly focus on light weight, low cost, and sustainable end products. These pressing constraints make the manufacturers consider the implementation of SM technologies. However, the barriers of implementation are in multiple categories. The need of initial cost of implementation, highly qualified workforce and construction/maintenance factors are becoming main issues for the manufacturers at the time of decision making.

In order to keep the manufacturing companies competitive and capitalize on new opportunities, developing and implementing a SM assessment tool can help the manufacturers prepare their operations for the challenges of today's technological advancements. Such an assessment tool could include and survey a number of key areas in manufacturing facilities reported through the coverage of this review paper. The findings of this kind of assessment usually report the strengths and weaknesses of a company's abilities and readiness to the digital manufacturing world.

## 4. Conclusions

SM is increasingly accepted and adopted by industry and academia; more fields and research areas have found that there is potential in adapting the smarter manufacturing standards to the larger systems. Moreover, the more traction that SM gains, the more research questions arise. Overall, SM has a big advantage in energy-saving compared to traditional manufacturing. Several research studies reported throughout the review showed that products made though SM technologies required 50% to 75% less energy than large-scale manufactured goods.

SM combines multiple technologies, including but not limited to CPS, IoT, robotics/automation, big data analytics, and cloud computing. At present, only a few of these technologies have matured, and most of them are still in the development period before large-scale adoption. Manufacturing companies are still in the early stages of data application. Most companies use the collected data to explain the rules and root causes of historical performance, instead of using the data for predictive analysis to support decision-making.

The purpose of AM is to optimize agile supply chains to achieve sustainable production. SM is not purely automated production. With the support of the Internet of Things, this is the key to production and collaboration with workers and robots based on market demand. Each of these technologies can reduce energy consumption to a certain extent for production. The integration of SM will further reduce the energy consumption of the production industry and contribute to world environmental protection. Today, SM is still in its early stages. Given the high level of interest in various industries, it is foreseeable that rapid development in this area will be achieved soon. Due to its interdisciplinary nature, advances in the field of basic research may find methods for industrial applications faster than in the past few years. For researchers who do not have much interaction with applied research in their field, this may be an opportunity to collaborate with researchers in related fields and industries and witness the application of their research in real life.

**Author Contributions:** Writing—original draft preparation, S.T., H.L., I.F., Y.Z., K.T., T.G. and B.A.; writing—review, I.F. and K.T.; editing, S.T., H.L. and I.F. All authors have read and agreed to the published version of the manuscript.

**Funding:** This research was funded by NATIONAL SCIENCE FOUNDATION Advanced Technological Education Program, grant number 1801120, entitled 'Smart Manufacturing for America's Revolutionizing Technological Transformation'.

**Conflicts of Interest:** The authors declare no conflict of interest.

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
