# Peer review of "The Influence of Smart Manufacturing towards Energy Conservation: A Review"

_technologies, doi:10.3390/technologies8020031_

Round 1

Reviewer 1 Report

The paper takes its foundation in smart manufacturing from the NIST definition. This is an accepted approach even though additive manufacturing has been given excessive importance in this. The Background and introduction if factually correct. A somewhat more critical approach in this section would be beneficial as it is at the moment essentially several statements stacked onto of each other. It would be interesting if the authors could be guided into the differences with the different definitions and its consequences for the analysis and later more on the implications for resource and energy consumption. This would aid the reader in interpretation and increase the quality of the paper.

Section 2 har plenty of interesting information and is stacked with information. In many cases, there are statements on improvements, but how this was achieved is sparsely described. When more detail is given as from line 488 and on it becomes hard to read. Many of the savings made were through material savings and lightweight components. The connection to "smart industry" is weak. It would here be interesting if the authors could systematise the different ways smart industry resulted in specific saving. This could be made through for instance 1) Innovative designs through "smart methods," 2) Energy savings through Production gains, 3) improved yields in processes through digital twins or sensors ... and so forth.

Discussions are too sparse and more quantitative conclusions would be required.

The paper warrants publication but need revisions with a more critical background elaborating the differences and consequences for the different definitions and approached by the different organisation. A bit more of analysis, with a classification of outcomes of the smart industry approach, with quantitative indication on the savings. Give this a structure.

Author Response

Point 1: The paper takes its foundation in smart manufacturing from the NIST definition. This is an accepted approach even though additive manufacturing has been given excessive importance in this. The Background and introduction if factually correct. A somewhat more critical approach in this section would be beneficial as it is at the moment essentially several statements stacked onto of each other. It would be interesting if the authors could be guided into the differences with the different definitions and its consequences for the analysis and later more on the implications for resource and energy consumption. This would aid the reader in interpretation and increase the quality of the paper.

Response 1:  Thank you for questioning the differences in our definitions and more impact of our definitions to energy consumption. In several places of our paper, we tried to touch base the differences. Even at the beginning we highlighted the traditional manufacturing versus SM in 1.3. Later on, we provided comparative highlights on Digital Thread and Digital Twin. Energy savings aspects were debriefed in 2.2.2. In 2.4.2, we specifically presented the energy saving case for Robots. In 1.2.2, we specifically, provided energy saving aspects of Cyber Physical Systems. The following statements were added in several places of the paper.

‘In all these technologies, it was proven that additive manufacturing can be an excellent solution for the production needs of any business with minimal energy consumption, and selecting the appropriate additive manufacturing technology for the customers’ requirements on the basis of complexity, material, and batch size is the key for the successful outcomes.’

‘With the modern innovations IA has offered in the fields of additive manufacturing, robotics, and digital technologies, energy companies are now exploring the possibilities of incorporating IA to increase the more efficient utilization of energy.’

‘SM and data analytics have a potential for today’s manufacturing industry to solve the energy efficiency concerns from the equipment level to the entire manufacturing plant. With intelligent communication systems they provide, it is proven that both can easily make manufacturing industries more productive, affordable, and competitive.’

Point 2: Section 2 har plenty of interesting information and is stacked with information. In many cases, there are statements on improvements, but how this was achieved is sparsely described. When more detail is given as from line 488 and on it becomes hard to read. Many of the savings made were through material savings and lightweight components. The connection to "smart industry" is weak. It would here be interesting if the authors could systematise the different ways smart industry resulted in specific saving. This could be made through for instance 1) Innovative designs through "smart methods," 2) Energy savings through Production gains, 3) improved yields in processes through digital twins or sensors ... and so forth.

Response 2: Authors would like to thank the reviewer for the concern provided by him/her. Authors did their best to provide more in-depth and specific examples in these three categories. See the following additions throughout the paper.

‘Overall, several of these reported studies provided valuable information resources about the benefits of having CPPSs in their overall production yield and energy saving through production gains.’

‘As it could easily be seen from these reported studies, adding the IIoTs can enable direct energy savings for the smart factory of today.’

‘Overall, the Digital Thread/Twin technology helps the manufacturers with the information needed to form intelligent solutions for the reduced use of energy. Several studies reported before provides best practices and case studies which could be adapted and implemented.’

‘Several research and case studies reported above proved that additively manufactured products required 50% to 75% less energy than large-scale manufactured goods.’

Point 3: Discussions are too sparse and more quantitative conclusions would be required.

Response 3: We appreciate the concern provided by the reviewer. However, the review paper prepared for the Technologies Journal does not have any goal to report any measurable finding. The scope of the paper is to draw a generic review on SM’s impact to energy saving and the author team believes that several solid examples and studies presented report most current knowledge blocks to clarify the objectives of the paper. The following statements were added.

‘It is very clear that manufacturing has a new face in 21st century. The traditional method of manufacturing practices does not provide as much benefit when compared to the smarter technologies available to companies. Today, customer requirements mostly focus on light weight, low cost, and sustainable end products. These pressing constraints make the manufacturers consider the implementation of SM technologies. However, the barriers of implementation are in multiple categories. The need of initial cost of implementation, highly qualified workforce and construction/maintenance factors are becoming main issues for the manufacturers at the time of decision making.’

‘Several research studies reported throughout the review showed that products made though SM technologies required 50% to 75% less energy than large-scale manufactured goods.’

Point 4: The paper warrants publication but need revisions with a more critical background elaborating the differences and consequences for the different definitions and approached by the different organisation. A bit more of analysis, with a classification of outcomes of the smart industry approach, with quantitative indication on the savings. Give this a structure.

Response 4: As indicated above several additions were made.

Reviewer 2 Report

  1. The image resolution is not perfect, especially fig. 2
  2. Section 1.4 introduces NIST. Are there any other government agencies performing the similar function all over the world?
  3. It will be great to include more details on how to establish the digital twin, as well as the scope and operations.
  4. Line 267, is “avail-able” a typo? Line 411, “be-coming”
  5. Figure 6 (it should be a table), ceramic powder is usually not used for DMLS, SLM, or EBM (line 441-443). Laser beam is not the only energy source for sheet lamination. Currently, many LOM based AM processes don’t use laser. There is also electron beam for DED AM process.
  6. Line 450, besides drop-on-demand, there is also continuous mode.
  7. In the discussion section, it will be great to discuss the people or engineer readiness as part of the smart manufacturing for the industry.

Author Response

Point 1: The image resolution is not perfect, especially fig. 2

Response 1: Thank you for reporting the low quality of Figure 2. Now, a higher resolution image was placed for Figure 2.

Point 2: Section 1.4 introduces NIST. Are there any other government agencies performing the similar function all over the world?

Response 2: This is a good point. Thank you. Many US based agencies developing standards are international in their functioning. As an example, International Organization for Standardization (ISO) develops and supports a high number of standardization works on related concepts from energy efficiency of heat pumps to additive manufacturing.

Point 3: It will be great to include more details on how to establish the digital twin, as well as the scope and operations.

Response 3: Thanks for this suggestion. More depth was included as given below.

‘With the advancements in IoT technology, the popularity of the digital twin is increasingly growing.  Now, a sensor could easily be attached to a physical object and operational data could be remotely collected precisely and then, the object could be smoothly controlled from its digital twin. Creating a digital twin could be divided into several steps. Selecting a technology that helps your need in real time data flow from the IoT device is the beginning step. The next step is your decision on the operation of your digital twin. In monitoring your system, it is your main task to track the operational performance. Overall, your implementations could start small, but grow over time.’

Point 4: Line 267, is “avail-able” a typo? Line 411, “be-coming”

Response 4: Thank you for pointing both grammar errors. Both words were corrected and saved.

Point 5: Figure 6 (it should be a table), ceramic powder is usually not used for DMLS, SLM, or EBM (line 441-443). Laser beam is not the only energy source for sheet lamination. Currently, many LOM based AM processes don’t use laser. There is also electron beam for DED AM process.

Response 5: This is a very good catch and list of suggestions. Thank you. The table has been updated with several new information. All the suggestions made by reviewer 2 were made. Figure 6 was relabelled as Table 1 now. The table is crafted by Yunbo Zhang. No copyright permission will be needed.

Point 6: Line 450, besides drop-on-demand, there is also continuous mode.

Response 6: Thank you for pointing this addition. We made the addition indicating that ‘the material jetting technology is also available in ‘continuous’ mode.’

Point 7: In the discussion section, it will be great to discuss the people or engineer readiness as part of the smart manufacturing for the industry.

Response 7: Thank you for suggesting to add such an important point. We added the following statement.

‘In order to keep the manufacturing companies competitive and capitalize on new opportunities, developing and implementing an SM Assessment tool help the manufacturers prepare their operations for the challenges of today’s technological advancements. Such an assessment tool could include and survey a number of key areas in manufacturing facilities reported through the coverage of this review paper. The findings of this kind of assessment usually report the strengths and weaknesses of a company’s abilities and readiness to digital manufacturing world.’

Reviewer 3 Report

The paper is a review work about several aspects of smart manufacturing. Therefore, it discusses its advantages in terms of energy-saving and production efficiency. The literature revision is sufficient, in fact 114 references are enough for a review paper. However, a major revision is requested because of the following main points:

  • The graphical quality (in particular the resolution) of each figure must be improved, it is quite poor. For this reason, the figures are very difficult to understand.
  • The section 3 about “Discussion” is too short for a review paper. It must be enlarged by giving further details
  • Read again the paper to check for residual typing errors.

Author Response

Point 1: The paper is a review work about several aspects of smart manufacturing. Therefore, it discusses its advantages in terms of energy-saving and production efficiency. The literature revision is sufficient, in fact 114 references are enough for a review paper. However, a major revision is requested because of the following main points:

The graphical quality (in particular the resolution) of each figure must be improved, it is quite poor. For this reason, the figures are very difficult to understand.

Response 1: Thanks for the nice statements you make about our review work. We appreciate that. Yes, author team believes that this paper will be a very impactful review for the researchers and engineers who are in the field of advanced manufacturing practices.

Figure 2 and Table 1 were remade. They were added to the revised paper. Attached are the extra copies.

Point 2: The section 3 about “Discussion” is too short for a review paper. It must be enlarged by giving further details

Response 2: Yes, further enhancements were made. Additions are at the end of discussions.

Point 3: Read again the paper to check for residual typing errors.

Response 3: Author team did read the paper one more time for any factual errors. Thanks so much for this advice.

Reviewer 4 Report

Dear authors,

The article gives a good review of the various kinds of smart manufacturing technologies and presents a reasonable set of references in documenting smart manufacturing technologies. The authors correctly point out in the abstract, that the article only 'introduces' the advantages in terms of energy-saving and production efficiency. However, it falls short of connecting the dots and providing a good discussion on the the influence of smart manufacturing on energy efficiency. 

A review paper typically focuses on the discussion, while the current work was focused on the materials and method section. Each of the sub-sections goes into a lot of detail in explaining the smart manufacturing technology, but only spends a fraction of space on the topic of Energy savings & efficiency. Several key numbers are cited across the text, but it falls short of giving the reader a good review. Further, some important papers specifically on Energy savings and efficiency are not cited, as far as I could go through the references.

Here below I have included some additional and recently published references for the authors perusal. I apologize if some of the articles below have already been cited.

1) Meng, Y., Yang, Y., Chung, H., Lee, P. H., & Shao, C. (2018). Enhancing sustainability and energy efficiency in smart factories: a review. Sustainability10(12), 4779.

2) Liu, Y., Zhang, Y., Ren, S., Yang, M., Wang, Y., & Huisingh, D. (2020). How can smart technologies contribute to sustainable product lifecycle management?. Journal of Cleaner Production249, 119423.

3) Supekar, S. D., Graziano, D. J., Riddle, M. E., Nimbalkar, S. U., Das, S., Shehabi, A., & Cresko, J. (2019). A Framework for Quantifying Energy and Productivity Benefits of Smart Manufacturing Technologies. Procedia CIRP80, 699-704.

4) Ren, S., Zhang, Y., Liu, Y., Sakao, T., Huisingh, D., & Almeida, C. M. (2019). A comprehensive review of big data analytics throughout product lifecycle to support sustainable smart manufacturing: A framework, challenges and future research directions. Journal of cleaner production210, 1343-1365.

Kind regards

Author Response

Point 1: The article gives a good review of the various kinds of smart manufacturing technologies and presents a reasonable set of references in documenting smart manufacturing technologies. The authors correctly point out in the abstract, that the article only 'introduces' the advantages in terms of energy-saving and production efficiency. However, it falls short of connecting the dots and providing a good discussion on the the influence of smart manufacturing on energy efficiency.

Response 1: Authors appreciate the concern provided by the reviewer. Throughout the paper, several additions were made to reflect the influence of smart manufacturing on energy efficiency.

Point 2: A review paper typically focuses on the discussion, while the current work was focused on the materials and method section. Each of the sub-sections goes into a lot of detail in explaining the smart manufacturing technology, but only spends a fraction of space on the topic of Energy savings & efficiency. Several key numbers are cited across the text, but it falls short of giving the reader a good review. Further, some important papers specifically on Energy savings and efficiency are not cited, as far as I could go through the references.

Here below I have included some additional and recently published references for the authors perusal. I apologize if some of the articles below have already been cited.

1) Meng, Y., Yang, Y., Chung, H., Lee, P. H., & Shao, C. (2018). Enhancing sustainability and energy efficiency in smart factories: a review. Sustainability, 10(12), 4779.

2) Liu, Y., Zhang, Y., Ren, S., Yang, M., Wang, Y., & Huisingh, D. (2020). How can smart technologies contribute to sustainable product lifecycle management?. Journal of Cleaner Production, 249, 119423.

3) Supekar, S. D., Graziano, D. J., Riddle, M. E., Nimbalkar, S. U., Das, S., Shehabi, A., & Cresko, J. (2019). A Framework for Quantifying Energy and Productivity Benefits of Smart Manufacturing Technologies. Procedia CIRP, 80, 699-704.

4) Ren, S., Zhang, Y., Liu, Y., Sakao, T., Huisingh, D., & Almeida, C. M. (2019). A comprehensive review of big data analytics throughout product lifecycle to support sustainable smart manufacturing: A framework, challenges and future research directions. Journal of cleaner production, 210, 1343-1365.

Response 2: The author team is against the issues raised by reviewer 3 although they greatly appreciate the statements made by him/her. As a team, we would like to justify our reasoning:

One: Corresponding author has just published a review paper almost a year ago. That paper has 23 citations in one year. Based on his former experience, the author guidelines for the Technologies Journal were taken as the key for the preparation of this manuscript. The team has followed the author guidelines provided at https://www.mdpi.com/authors/layout. As it could be seen from this website, there is no such rule indicating that the review paper will be focused on the discussions. The author team has made several enhancements and they will not be able to restructure paper and make the paper sub-sectioned under Discussion since it will require substantial structural changes in editing.

Two: After several months of hard works and reviews, the author team has reviewed 114 technical publications. Several of them are extremely new and provide in-depth and most recent research studies and technological advancements. Author team agrees not to add any further references since they all believe that the coverage of this manuscript is most up-to-dated, most advanced and ready to fill the gap for the engineers, scientists and researchers who need to learn the latest trends and technologies in SM. As it could be seen from the following six cited references, the team’s decision is not to any more concept and reflections from other studies provided by reviewer 3.

  1. S. Terry, I. Fidan, Y. Zhang, and K. Tantawi, ‘’Smart Manufacturing for Energy Conservation and Savings,’’ 2019 NSF ATE Principal Investigators Conference, https://par.nsf.gov/biblio/10140480.

  1. G. May, I. Barletta, B. Stahl, and M. Taisch, “Energy management in production: A novel method to develop key performance indicators for improving energy efficiency,” Appl. Energy, vol. 149, no. JULY, pp. 46–61, 2015.

  1. N. Mohamed, J. Al-Jaroodi, and S. Lazarova-Molnar, “Leveraging the Capabilities of Industry 4.0 for Improving Energy Efficiency in Smart Factories,” IEEE Access, vol. 7, pp. 18008–18020, 2019.

  1. S. Nimbalkar et al., “Smart Manufacturing Technologies and Data Analytics for Improving Energy Efficiency in Industrial Energy Systems,” ACEEE Summer Study Energy Effic. Ind., pp. 114–126, 2017.

  1. E. A. Rogers and A. Council, “How Smart Manufacturing Saves Money Smart Manufacturing Technologies and Energy Savings,” ACEEE, pp. 1–11, 2015.

  1. B. Kennell, “Smart Manufacturing: A Path to Profitable Growth,” Huffpost, 2015. [Online]. Available: https://www.huffpost.com/entry/smart-manufacturing-a-pat_b_7314828.

Issues with Figures 2, 3, and 6: There are three figures in your submission come from other references. We need you to seek for copyright permission for

reusing them.

Response to issues with Figures 2, 3, and 6: All three figures were made by the author team. They were designed and made by the members of author team and references were added to show the in-depth source of the graphics and information presented so that any reader can go back and see the knowledge blocks of these graphs. Let us just briefly explain the case of Figure 3. Figure 2 and 6 are simply the graphical representation of the knowledge blocks presented at the paper.

Figure 3 is one that our first author, Shane Terry created himself with only the icons coming from a source outside. They are all listed a free license with attribution and freepiks redirects to flaticon as an associate. Everything else is his own: the on/off switches above the dials, the editing of the microchips, the text boxes, and text.

https://www.flaticon.com/free-icon/thermostat_31029

https://www.flaticon.com/free-icon/code-rate_103065

https://www.flaticon.com/free-icon/switch-off_32137

https://www.flaticon.com/free-icon/computer-microprocessor_18089

https://www.flaticon.com/free-icon/computer-cpu_63990

Round 2

Reviewer 3 Report

The paper has been revised in accordance with the reviewer's suggestions. It is now accepted for publication.

Author Response

Thank you. Attached is the latest copy enhanced and revised by the author team.

Reviewer 4 Report

Significant revisions have been made and the comments were carefully considered. I would advise switching some of the discussion from the initial sections to make a more comprehensive discussion section. Given the changes, and the response to the concerns raised, I would be willing to accept the article for publication.

Author Response

There is no other discussion made before Section 3.

The new 3.0 Discussion made the summarized comprehensive discussion per initial requests.
